# Towards a Comprehensive Account of Rhythm Processing Issues in Developmental Dyslexia

**DOI:** 10.3390/brainsci11101303

**Published:** 2021-09-30

**Authors:** Tamara Rathcke, Chia-Yuan Lin

**Affiliations:** 1Department of Linguistics, Faculty of Humanities, University of Konstanz, 78464 Konstanz, Germany; 2Modern Languages and Linguistics, School of Cultures and Languages, University of Kent, Canterbury CT2 7NR, UK; c.lin@kent.ac.uk; 3Department of Psychology, School of Humanities and Health Sciences, University of Huddersfield, Huddersfield HD1 3DH, UK

**Keywords:** speech rhythm, developmental dyslexia, phonological awareness, sensorimotor synchronization, phoneme monitoring, rhythm perception, motor entrainment

## Abstract

Developmental dyslexia is typically defined as a difficulty with an individual’s command of written language, arising from deficits in phonological awareness. However, motor entrainment difficulties in non-linguistic synchronization and time-keeping tasks have also been reported. Such findings gave rise to proposals of an underlying rhythm processing deficit in dyslexia, even though to date, evidence for impaired motor entrainment with the rhythm of natural speech is rather scarce, and the role of speech rhythm in phonological awareness is unclear. The present study aimed to fill these gaps. Dyslexic adults and age-matched control participants with variable levels of previous music training completed a series of experimental tasks assessing phoneme processing, rhythm perception, and motor entrainment abilities. In a rhythm entrainment task, participants tapped along to the perceived beat of natural spoken sentences. In a phoneme processing task, participants monitored for sonorant and obstruent phonemes embedded in nonsense strings. Individual sensorimotor skills were assessed using a number of screening tests. The results lacked evidence for a motor impairment or a general motor entrainment difficulty in dyslexia, at least among adult participants of the study. Instead, the results showed that the participants’ performance in the phonemic task was predictive of their performance in the rhythmic task, but not vice versa, suggesting that atypical rhythm processing in dyslexia may be the consequence, but not the cause, of dyslexic difficulties with phoneme-level encoding. No evidence for a deficit in the entrainment to the syllable rate in dyslexic adults was found. Rather, metrically weak syllables were significantly less often at the center of rhythmic attention in dyslexic adults as compared to neurotypical controls, with an increased tendency in musically trained participants. This finding could not be explained by an auditory deficit in the processing of acoustic-prosodic cues to the rhythm structure, but it is likely to be related to the well-documented auditory short-term memory issue in dyslexia.

## 1. Introduction

### 1.1. The Role of Phonology in Dyslexia

Dyslexia is considered a neurodevelopmental disorder [1], typically characterized by difficulties with an individual’s command of written language [2,3,4,5,6]. The issues with reading, (hand-)writing and spelling cannot be accounted for by an intellectual deficit [7], though other linguistic abilities may also be delayed in children with dyslexia [8]. The delays concern a slower speech production development (specifically the pronunciation of longer, more complex words), as compared to age-matched children without reading or writing difficulties [3]. The native language of the child is known to influence the severity of the disorder [9]. Less severe reading problems occur amongst children whose native languages have a more transparent grapheme-to-phoneme mapping [10]. Conversely, languages with a less transparent grapheme-to-phoneme mapping (such as English) pose more problems for children with dyslexia, and lead to more severe cases [2,9,10].

Whether or not a phonological deficit might be the underlying issue in dyslexia has been a matter of long-standing debates [11,12,13,14], with a large body of evidence documenting a tight relationship between phonological awareness and reading ability [15,16,17,18]. ‘Phonological awareness’ is an umbrella term used to describe different aspects of auditory processing of structural phonological units—such as phonemes, syllables, words or sentences—within one’s native or ambient language [19,20,21,22], and is thought to lay a crucial foundation to the general language processing system [23]. A well-developed phonological awareness involves an efficient encoding of an entire phonological system and indicates that an individual has the ability to recognize, discriminate, retain and manipulate the sound structure of their language(s) [22,24]. This awareness has been widely reported as impaired in dyslexic listeners, as indicated by a reduced ability to segment lexical items into discrete phonological units such as syllables and phonemes [9,25,26], poor phoneme awareness [27,28], a reduced short-term memory capacity for verbal information and a slow retrieval of lexical information [14,29]. Overall, the deficit of input representations is assumed to be more severe than the deficit of phonological output representations [29].

The mapping of variable acoustic signals onto discrete phonological representations of the listener’s native language(s) is one of the major challenges in speech perception [30]. Children with dyslexia seem to have a deficit in perceiving speech sounds in a categorical fashion like their age-matched or reading-level matched peers, i.e., they tend to make insufficient use of acoustic cues to phonemic contrasts while extensively attending to the acoustics of allophonic variation [31,32,33,34], (see [35] for a meta-analysis). Such allophonic perception is not specific to dyslexia. It has generally been documented in preschoolers and is known to give way to categorical perception with increased reading experience as the schooling progresses [36], though at the physiological level of neural activation, the tendency to allophonic (instead of categorical) perception may persist even in well-compensated adults with developmental dyslexia [37]. 

### 1.2. Motor Entrainment Deficits in Dyslexia

Individuals diagnosed with developmental dyslexia have shown a number of motor deficits indicative of issues with rhythmic entrainment and potentially suggestive of an underlying rhythm processing impairment. For example, Wolff [38] applied a finger tapping paradigm to the study of rhythmic sensorimotor synchronization (SMS) abilities in dyslexia. Twelve adolescents diagnosed with developmental dyslexia were asked to tap in time with auditory prompts of varied complexity and at different time scales. The results demonstrated that in comparison to a neurotypical control group, dyslexic participants tended to produce larger negative mean asynchronies while synchronizing with a metronome, and experienced increased difficulties with the phase correction when tapping to a metronome as well as with the motor reproduction when exposed to simple auditory rhythms. Wolff et al. [39] tested dyslexic and control participants from two age groups—adolescents and young adults—and asked them to perform a series of non-linguistic SMS tasks. Accordingly, the study groups differed in their tapping stability, with dyslexic adolescents showing greater motor instability than all other participants. Thomson et al. [40] investigated the SMS performance of dyslexic adults and used a finger tapping paradigm following the methodology by [38,39], and additionally including an unpaced tapping task that measured participants’ spontaneous tapping rate. No group differences were found for negative mean asynchronies with a metronome (contra [38]). Instead, dyslexic adults showed greater variability of their inter-tap intervals during both paced and unpaced tapping. This finding was only partially in line with the results discussed in Wolff et al. [39] who identified a significant tapping instability for the adolescent but not the adult group of dyslexic participants. 

More broadly, motor synchronization skills seem to be interrelated with the acquisition of literacy. As shown by Tierney and Kraus [41], individual variability in tapping with a metronome correlates with reading skills in normally developing adolescents. However, previous results based on dyslexic participants’ performance with non-linguistic stimuli have provided somewhat inconclusive evidence for a generally impaired motor entrainment which might be indicative of the fact that dyslexia is primarily a language-specific impairment [42]. Thus, the involved rhythm processing difficulties might be more readily amenable to an empirical observation in a linguistic rather than a non-linguistic setting. To date, only a limited number of studies have employed SMS to investigate dyslexic listeners’ perception of rhythm in spoken language. Wood and Terrel [43] asked children with and without a reading impairment to tap or clap along with the syllables in individual spoken words, as a means of counting how many syllables each word contained. Results showed that among three groups of readers with different abilities (poor readers, age-matched controls and reading age controls), poor readers scored low while the age-matched controls achieved the highest scores for their ability to correctly clap out the number of syllables in a word; however, the group difference in the scores was not significant. 

Leong and Goswami [22] studied SMS with short, metrically regular speech in dyslexic and neurotypical adults. The experimental sentences of the study included four traditional nursery rhymes that contained either a trochee (SW, e.g., ‘Mary Mary quite contrary’) or an iamb (WS, e.g., ‘The Queen of Hearts she made some tarts’). Each rhyme was presented three times (with short silent breaks in-between) while participants tapped along to the perceived rhythm of each rhyme, maintaining their tapping during the silent intervals between repetitions. The results showed that neither the inter-tap interval (ITI) duration nor the asynchronies measured with the onset of lexically stressed vowels (a proxy of syllables’ perceptual centers, [44,45]) differed across the two groups of participants. Instead, Leong and Goswami [22] argue that the SMS performance of the two participant groups could be best described as divergent at the level of the syllabic entrainment while the stress- and phoneme-level entrainment was comparable in participants with dyslexia and the neurotypical controls. The conclusion was based on the acoustic analyses of amplitude envelopes obtained from the nursery rhymes, and a subsequent decomposition of amplitude modulations into instantaneous frequencies using a Hilbert transform ([46], cf. [47]). Such analyses aim at representing speech at multiple timescales: thought to correspond to phoneme, syllable and stress fluctuations inherent to spoken language. Following a signal decomposition, the accompanying SMS behaviors can be then described as being variably phase-locked to the different timescales of speech ([22], cf. [48]).

As our research has shown [48], signal decomposition and its derivatives do not represent consistent SMS anchors in speech. In comparison to manually identified vowel onsets, any amplitude derivative is significantly worse at predicting the location of finger taps collected during SMS with natural speech, even in healthy participants. In contrast, vowel onsets constitute stable SMS anchors in simple verbal stimuli [49] as well as in complex natural sentences [48,50]. While decompositional approaches seem to excel at tracking temporal fluctuations pertaining to different layers of prosodic hierarchy and occurring within large timescales of longer speech samples [47], the local time resolution of such approaches is, however, relatively poor, but is ultimately required for an adequate temporal representation of perceptually relevant rhythmic events (cf. [44,45]).

An alternative approach to the study of SMS with natural speech was developed in our previous research [49,50]. It combines the necessary temporal precision at the local, intra-syllabic level [44,45] with the hierarchical metrical representation of linguistic units [51,52,53]. Accordingly, SMS is collected while participants synchronize with what they perceive as the beat of sentences played back to them on a loop (e.g., 20, not just three, repetitions of a sentence as in previous work [22,54]). Individual and group SMS performance is then described as probability functions for the time course of a sentence, specifying local time points that are most/least likely to attract a tap and comparing them to the metrical representation of the sentence. We have repeatedly demonstrated that vowel onsets serve as consistent anchors of SMS in English [48,50], and that SMS-likelihood is also shaped by the properties of the prosodic hierarchy. Vowels in metrically strong positions are more likely to attract a finger tap than vowels in metrically weak positions, at least in English [50]. Moreover, our approach allows for the calculation of SMS accuracy which we also showed to be influenced by the metrical weight of a vowel within the prosodic hierarchy: English participants tend to tap more precisely with metrically strong than with metrically weak vowels [50]. Overall, the results obtained with our version of the SMS paradigm resemble a series of well-established SMS findings collected for temporally more regular types of auditory signals, such as metronome and music (see [55,56] for an overview). Importantly, individual variations in SMS can be quite large (cf. [57]) and are known to be influenced by the degree of musical training of the participants ([50] cf. [58,59]).

The study of the effects that music training has on the human brain and cognitive development is a burgeoning field (see [60] for an overview of the key findings). Musical training is often seen as “a resource that tones the brain for auditory fitness” [60], with a number of auditory perception benefits that extend to language. For example, music training is related to an improved speech perception in challenging listening conditions [61,62,63], increased verbal working memory [61,64,65,66] and enhanced speech segmentation skills [67,68]. Even non-professional musicians show such benefits (e.g., [69,70]). As far as rhythm perception and time-keeping skills are concerned, musically trained non-expert participants show fewer errors and less variability when synchronizing with a metronome than musically untrained participants (e.g., [71,72,73,74]). Our previous work has shown that these abilities also transfer to SMS with language [50]. Accordingly, participants with higher levels of musical training (which included playing an instrument, singing and dancing) also produced smaller asynchronies when synchronizing with natural speech.

The idea that musical training might be associated with an enhancement of reading abilities during childhood has been investigated in a number of studies (e.g., [75,76,77,78,79]). Music-based interventions in (pre-)schoolers have also been tested as a means of mitigating and treating developmental dyslexia (e.g., [80,81,82,83,84]). Even though not all studies have been able to demonstrate that music training leads to a significant improvement of individual difficulties with reading and writing, a major gain in phonological awareness skills has been documented in most previous research (see [85] for a meta-analysis). The facilitating effect of music training on phonological awareness has been demonstrated even before the onset of literacy acquisition, in 3 y.o. toddlers [84]. Improved phonological awareness is thought to moderate the impact of music training on literacy development, via changes in auditory processing mechanisms during music practice (e.g., [60,85,86,87]). Interestingly, a recent study with adult dyslexic and control participants showed that individual differences in musical rhythm ability captured the variability in perception of rhythmic grouping of spoken units better than the individual’s dyslexia status did [88].

### 1.3. Processing of Rhythm and Its Acoustic Correlates as the Underlying Issue in Dyslexia

Auditory and motor systems are known to be tightly connected during the processing of rhythmic patterns across a diverse range of acoustic signals (e.g., [89,90,91,92,93,94,95]). Given the importance that rhythm, or more broadly prosody, plays in speech processing, dyslexic difficulties might stem from the insufficient processing of prosody and its acoustic correlates. Prosody is known to guide speech segmentation at various levels of linguistic structure, including phonemes [96,97,98,99,100], morphemes [101] and words [102,103,104,105]. In a language like English, speech segmentation is facilitated by the presence of lexical stress which directs perceptual attention to potential word onsets [106,107], given that word-initial stress prevails in the English lexicon [108]. Alternations of strong and weak syllables cuing the distribution of lexically stressed syllables in sentences create unique rhythmic templates that shape spoken word recognition and facilitate lexical access [109]. Individual sensitivity to speech prosody is known to correlate with reading abilities and the acquisition of phonology [22]. In particular, sensitivity to rhythmic alternations of strong (S) and weak (W) syllables that encode metrical representations of words and sentences shows processing deficits in individuals with dyslexia as compared to age-matched controls [22,43,110,111,112,113,114,115].

A series of production and perception studies have produced compelling evidence that dyslexic individuals (both children and adults) do not efficiently encode metrical templates in English. For example, Goswami et al. [110] implemented a ‘Dee Dee’ task [114,116] and asked 12-year-old children to replace syllables of highly recognizable personal names with a metrical template consisting of strong and weak versions of the syllable ‘dee’ (e.g., ‘Harry Potter’ would be ‘DEEdee-DEEdee’). The results showed that dyslexic children performed poorly in this production task, compared to their age-matched controls. In a perceptual discrimination task, Leong and colleagues [115] presented their dyslexic participants with a series of English four-syllable words that were produced with either a SWWW or a WSWW pattern (e.g., MAternity or maTERnity, DIfficulty or diFFIculty) and then paired to either have the same or a different stress pattern. Compared to age-matched neurotypical controls, adult participants with diagnosed developmental dyslexia were significantly less accurate in judging stress templates across pairs of words, irrespective of the SWWW or WSWW stress judgement required or the same/different lexical pairings involved.

The question of whether or not an impairment of basic auditory processing mechanisms might cause such deficits in individuals with dyslexia, and which specific acoustic properties of speech signals might be processed insufficiently, has been addressed in a number of studies (see [117] for a review). Prosodically relevant acoustic cues include the fundamental frequency (F0), duration and intensity (e.g., [118,119]), though lexical stress in English relies less on F0 [120,121] and more on vowel quality and the corresponding formant frequencies [122,123]. Interestingly, the perception of formant frequencies does not seem to cause any difficulties to dyslexic participants [124,125]. In a perception study with manipulated format trajectories that distinguished between /ba/ and /wa/, Goswami et al. [125] found that dyslexic children had an increased sensitivity to the changes in vowel formants which encode /ba/ vs. /wa/, and in fact outperformed the age-matched controls on this task. The finding points toward a specific auditory processing strength in dyslexic listeners [124,125], and is somewhat at odds with their weak perception of lexical stress, given that one of its main phonetic correlates is vowel quality which is encoded by formant frequencies ([122,123], cf. [120] for a cross-linguistic review).

In a series of experiments, Goswami and colleagues have demonstrated that the individual sensitivity to stress patterns (d’) is correlated with the sensitivity to acoustic signal rise-time, but not with the sensitivity to signal intensity or frequency (e.g., [110,115,125,126]). Similarly, a meta-analysis by Hämäläinen et al. [117] reported that all previous experiments that tested rise-time perception in dyslexic participants identified a relationship between individual rise-time sensitivity and reading ability. In contrast, existing findings with regards to the role of other prosodically relevant cues such as duration, frequency and intensity are rather mixed [117]. According to some proposals (e.g., [127,128,129]), an efficient perceptual encoding of the acoustic properties of signal rise-time is key to the perception of metrical structure in speech, and prosodic processing difficulties observed in dyslexic listeners arise as a consequence of a lack of such encoding. However, previous research into the relationship between acoustic signal rise-time and metrical encoding difficulties in dyslexia has employed purely psychoacoustic tests to measure rise-time sensitivity in listeners (e.g., [40,110,115,129,130,131]) or perception tests involving simple /ba/-/wa/ monosyllables [125]. In natural speech, the role of syllable rise-time for the perception of rhythm is less clear-cut, and existing evidence is less conclusive ([48] cf. [132]).

Importantly, rise-time modulations depend on a large number of both acoustic and linguistic factors in natural connected speech. The duration of the syllable rise-time co-varies with its duration and intensity [48], and encodes a number of linguistic functions beyond prosody, including the manner of consonant articulation [133,134] and complexity of syllable onsets [48]. Given this polyfunctionality of amplitude rise-time modulations, linguistic consequences of an impaired rise-time perception cannot be clearly defined, thus impeding causality in the discussion of the underpinnings of dyslexia. 

### 1.4. Open Questions and Hypotheses

The main aim of the present study is to investigate rhythm perception and motor entrainment in dyslexia using natural speech, and to deepen the understanding of the underlying deficits in adults with dyslexia.

Most of the existing research that documented metrical encoding difficulties in dyslexia tested rhythm perception in simple words and phrases (see [117] for an overview) or studied the motor entrainment to metrically regular nursery rhymes [22]. However, natural speech is syntactically complex and lacks such regularities of strong and weak syllables, or any acoustic isochrony to support the temporal prediction of rhythmically relevant events [135,136]. Thus, we can expect natural speech to cause more difficulties to prosodic processing in dyslexic participants. To study rhythm perception and entrainment in complex auditory signals such as natural speech, the present study implements an SMS paradigm that we developed previously [50]. In the methodologically motivated study [50], we investigated SMS with looped spoken sentences and tested several acoustic anchors that might attract SMS in natural speech. Our findings showed that spoken sentences can reliably entrain movement when looped, and that (neurotypical) participants move in synchrony with vowel onsets when asked to synchronize to the sentence beat. Crucially, the paradigm is sensitive to the metrical structure of sentences as strong syllables tend to evoke more taps and increase the synchronization accuracy. The present research applies the established paradigm to test rhythm perception and entrainment in dyslexia, and uses a larger number of sentences with syllabic nuclei being occupied not only by vowels but also by consonants. Nasals and laterals can constitute the nuclei of weak syllables in natural speech in English [137]. These sounds are relatively high on the sonority scale and are acoustically distinguished by the presence of a formant structure similar to vowels, though with an overall lower amplitude [138,139,140]. 

If dyslexic listeners have rhythm processing difficulties due to their issues with the syllabic entrainment as Leong and Goswami [22] suggest, we would expect dyslexic participants to produce fewer and less precise taps for all syllables, regardless of the syllable’s metrical weight. If the rhythm processing difficulty arises from an auditory issue with the encoding of lexical stress and the hierarchical relationships between stressed and unstressed syllables, we would expect adult participants with dyslexia to perform worse than the age-matched controls in their SMS with the vowels of strong syllables only which would be indicative of an entrainment deficit with longer time-scales. 

Moreover, the role of prosody in phonological awareness is unclear. Previous research has not conclusively demonstrated if, and how, the ability to segment and identify concrete phonemes in variable, continuous acoustic speech signals is interrelated with the ability to perceive metrical hierarchies of lexical and phrasal prominences. The two aspects of auditory speech processing can be linked in three ways. First, prosody can support the bootstrapping of segmental phonology in natural speech (cf. [141]), and thus be the origin of the dyslexic deficits reported above [22,43,110,111,112,113,114,115,125]. Second, the inability to perceive metrical relationships between strong and weak syllables might be a consequence of a limited ability to attend to the relevant acoustic cues encoding segmental contrasts in connected speech. The third possibility is that the two aspects of auditory speech processing play an independent role in phonological awareness, and their deficits are unrelated in dyslexia. To test the nature of the relationship between phoneme awareness and rhythmic skills, we additionally asked our participants to perform a phoneme monitoring task by listening to nonsense strings containing target obstruents and sonorants. Phoneme monitoring taps phonological awareness by testing the ability to attend to variable acoustic cues, to segment a continuous speech stream into phoneme-sized units, and to map those units onto corresponding phonological representations. Since nonsense strings contain no real words that could facilitate phoneme access via lexical boost, the task’s demands are purely acoustic. Given that consonant contrasts encoded in formant dynamics rarely cause processing issues for dyslexic listeners [124,125], we expected dyslexic listeners to show little difficulties in identifying sonorants but perform worse than the control participants (i.e., with lower accuracy and longer reaction times) when monitoring for obstruents (cf. [35,142]).

To understand the relationship between phoneme perception and prosodic processing, participants’ performance in each task is used as a predictor of their performance in the other task. That is, individual sensitivity to variable acoustic representations of target phonemes embedded in nonsense strings (d’) is fit as a predictor of participants’ SMS, and vice versa, individual SMS rates with strong and weak syllables in the SMS task are fit as predictors of participants’ phoneme monitoring performance. Depending on the predictive power of these individual measures in the best-fit models, conclusions can be drawn on the relationships between phoneme perception and rhythm processing.

Moreover, whether or not dyslexia results from a general auditory deficit of linguistically relevant aspects of signal acoustics has been controversially debated (see [132] for an overview). Previous research into the underlying causes of the deficits in phonological awareness and prosodic processing has frequently relied on correlational data obtained from substantially different types of tasks, both linguistic and non-linguistic. Therefore, the discussion of causality among the involved processes and the scope of the derived conclusions are to be considered limited (cf. [132]). Studies of the auditory processing deficits in dyslexia have mostly employed psychoacoustic tests to measure individual sensitivity to the acoustic properties of rise-time, duration or intensity, without attesting their direct links to the prosodic and segmental categories at hand [40,110,115,129,130,131]. However, acoustic cues such as rise-time, duration and intensity encode a multitude of linguistic functions and categories in connected speech [133,143,144,145]. If an auditory processing deficit of (any of) these acoustic parameters was at the heart of dyslexia, their impact should be measurable, especially in verbal tasks. Therefore, the present study seeks to estimate the effects of variable acoustic rise-time, duration and intensity that are simultaneously present in the stimuli, on participants’ performance in the two experimental tasks: phoneme monitoring and SMS.

Finally, given that music practice changes general auditory processing mechanisms that are also pertinent to speech and language (e.g., [60,85,86,87]), we expect levels of musical training to play a role in the two tasks. Musically trained participants, regardless of their dyslexia status, are likely to show better performance in the experiments (i.e., faster RTs, lower asynchronies, higher accuracy and SMS rates), potentially highlighting the higher importance of musical training over dyslexic difficulties among well-compensated adults [88].

## 2. Methods

### 2.1. Materials and the Phoneme Monitoring Task

The stimuli for the phoneme monitoring task were selected from the materials used in previous research [105]. The materials of this study comprised short nonsense sequences (e.g., 3-syllable [bləˈkinɪm] or 4-syllable [ʃɪkləˈtiðɪʒ]) whose metrical structure was either WSW or SWSW. In all sequences, the penultimate syllable bore primary stress, and the weak vowel quality was either a schwa or a high front lax vowel [ɪ]. The original study implemented a word-spotting paradigm to investigate the role of cross-dialectal timing cues in lexical segmentation and access. However, the task turned out to be quite difficult for listeners, with average spotting rates around 30% or below.

For the purposes of the present study, 40 experimental stimuli and 40 distractors were identified in the materials. The target phonemes contained either an obstruent (/p/, /k/), or a sonorant (/m/, /l/). The distractor sequences did not contain the target phonemes. The 80 stimuli were grouped into eight blocks of 10, each comprising of five targets and five distractors in a random order. Each target phoneme was monitored for in a total of two blocks. The order of testing blocks was pseudo-randomized, prohibiting one phoneme from being the target of two consecutive blocks. A practice block containing five /s/- targets and five non-targets was given before a formal testing session.

At the start of each block, participants were instructed to monitor for one of the target phonemes and tap on a tambourine as soon as they heard the specified target. The tambourine was placed at a comfortable distance in front of the participants. It was connected to a piezoelectric sensor. An Arduino microcontroller linked the sensor to a Dell Latitude 7390 laptop running the PsychoPy software which recorded and extracted the timing of participants’ taps. 

The rise-time, duration and intensity were measured for each target. The rise-time of the syllables containing the target was calculated following the same routines that we employed in our previous research [50]. Accordingly, amplitude envelopes were derived by the *envelope* function in Matlab (version 9.5 (R2018b), The MathWorks Inc., Natick, MA, USA), which operated on the absolute signal amplitude, smoothing it by applying a spline interpolation with an 11 ms window. The rise-time measured the temporal distance between a local amplitude minimum and a maximum identified in the smoothed envelope contour around each syllable onset [21,129]. The duration and the intensity were measured from sentence spectrograms in Praat [146]. The duration of stops included both the closure and the VOT (e.g., [147]). The average intensity of syllables containing targets was normalized with reference to the mean intensity of the whole sentence. These measurements are summarized in Table 1, along with the results of an analysis of variance testing whether the acoustic difference between the obstruent and the sonorant targets was consistent and significant in these stimuli.

To capture individual differences in the sensitivity to the acoustic cues of the target’s phonemes embedded in a nonsense string, individual d’-values (MacMillan and Creelman, 2005) were calculated for each participant and used as a predictor in the SMS models. All target phonemes were included in the d’-calculation, given that no significant differences in spotting sonorants vs. obstruents were observed across the two groups of participants (see Section 3.2).

### 2.2. Materials and the SMS Task

Twenty English sentences were spoken by a 30-year-old female native speaker of Standard British English and recorded at 44,100 Hz sampling rate in a sound-attenuated booth. The sentences varied in the overall number of syllables (ranging from 7 to 13) and their duration (ranging from 1.2 to 2.3 s). Each sentence contained one or two syllables with sonorant nuclei (see Appendix A for the full sentence list). For each syllable, rise-time, nucleus duration and intensity were measured as described above. Similar to the preparation of the materials for the phoneme monitoring task, the average intensity of each nucleus was normalized with reference to the mean intensity of the corresponding test sentence. The acoustic measurements are summarized in Table 2, broken down by the metrical weight of the syllable containing the corresponding nucleus (strong vs. weak) and the type of the nucleus in the weak syllable (vowel vs. sonorant).

Sentence loops were then created, following previously established procedures of measuring SMS with natural linguistic stimuli [50]. Accordingly, each sentence was repeated 10 times, with a 400 ms silent pause between repetitions. The order of the experimental sentences was randomly created for each individual session. Participants were instructed to start synchronizing the movement of the index finger of their dominant hand in time with the beat of the spoken sentence from its third repetition until the end of the loop. Please note that participants were not explicitly advised to synchronize their taps with every syllable nucleus (though this tapping pattern best reflects the subjective experience of the speech beat that we previously documented in neurotypical participants using the SMS task, [50]). 

Prior to the formal testing, participants had an opportunity to practice their understanding of the SMS task with five sentence loops containing 20 repetitions and no instructions given as to when to start synchronizing. Participants’ finger taps were recorded on the tambourine that was also used in all other tasks throughout the experiment. 

To capture the individual differences in the entrainment to syllable nuclei, individual SMS rates were calculated as the number of syllables with which a participant synchronized, divided by the total number of syllables in the materials. If, in the course of sentence repetitions, participants did not consistently produce a tap within a ±120 ms window around a nucleus onset [50,55], the syllable was identified as lacking SMS (e.g., the third syllable in the example shown in Figure 1). Thus, the SMS rate expressed the proportion of nuclei with which a participant synchronized, and was fit as a predictor in the models of the phoneme monitoring data.

### 2.3. Other Motor Tasks

General individual SMS abilities were captured by four additional motor tasks. In two unpaced tapping tasks, participants were instructed to tap at their most comfortable speed for one minute, and then to tap at their fastest possible speed for 30 s. No auditory prompts were given in the unpaced tapping tasks. 

In two paced tapping tasks, participants were instructed to tap in synchrony with two auditory sequences consisting of 25 repetitions of the syllable [bi:] (“bee”, cf. [49]). The first sequence contained the syllable presented at an inter-onset-interval (IOI) of 600 ms while the second sequence had an IOI of 300 ms. Note that instead of a metronome or pure tones, we used a simple verbal prompt in the paced tapping task as our previous research demonstrated that the SMS performance in the paced tapping task is comparable across a range of verbal and non-verbal stimuli as long as they share the temporal frame of occurrence ([49], see also [135]).

### 2.4. Experimental Procedure 

Participants were invited to a quiet room of the Linguistics Laboratory at the University of Kent and gave written consent prior to taking part in the experiment. At the beginning of an experimental session, they completed a short 5-item musicality questionnaire. Numerical codes were assigned to each answer and summed up in an individual musicality score ([50], cf. [148]). The questionnaire asked participants to self-report whether they had any musical training (0 for “no”, 1 for “yes”) and were still actively practicing their hobby (0 for “no”, 1 for “yes”). Further questions collected information about the age they started their training (2 for 0–10 y.o., 1 for 10–20 y.o., 0 for 20+ y.o.), the number of years they engaged with playing an instrument or singing (0–14 in the present sample), and how many instruments they played (0–4 in the sample, including singing). The derived musicality score was a numerical composite of the questionnaire answers, with higher scores indicating a higher level of musical training and experience ([50], cf. [148]).

Upon completion of the questionnaires, participants engaged with a series of finger-tapping tasks presented to them in a fixed order, consisting of (1) SMS with the linguistic stimuli, (2) two unpaced tapping tasks, (3) two paced tapping tasks, one at an IOI of 600 ms and one at an IOI of 300 ms. An experimental session concluded with the phoneme monitoring task, followed by a standardized 15-item checklist screening for dyslexia [149]. The playback of all experimental stimuli and the recording of participants’ taps were administered in PsychoPy. Good quality headphones (Sennheiser HD 380) were used in the experiment, with participants being free to adjust the volume to an individually comfortable level. The whole experiment took approximately 40 min to complete. 

### 2.5. Participants

Twenty-five adults (6 M) participated in the current study. Attention was paid to matching the age and the gender of dyslexic and control participants during recruitment. All participants were British monolinguals without any known hearing or motor disabilities. All participants gave informed written consent to participate in the experiments, and were compensated for their time. The study received ethical approval (reference number: 0291920, received 6 January 2020) from the Faculty of Humanities Research Ethics Advisory Group for Human Participants at the University of Kent.

One female participant was excluded from further analyses because her taps were too soft to be detected by the tambourine. Thirteen (of the remaining twenty-four) participants self-reported to have been officially diagnosed with developmental dyslexia (3 M, mean age = 21.5 years, SD = 1.2 years) while eleven participants self-reported as having normal reading and writing abilities (3 M, mean age = 21.6 years, SD = 1.2 years). Individual dyslexia scores were calculated for each participant based on their answers to a standardized checklist [149]. Previous research established that the scores obtained from the adult checklist were correlated with objective measures of literacy skills [150]. As expected, the control group had a low average score indicative of no dyslexic difficulties (mean: 35.0, *SD*: 3.9) while the dyslexic group had a higher average score consistent with symptoms of mild dyslexia (mean: 58.2, *SD*: 9.5; Welch’s two sample *t*-test: *t*(16.7) = 8.14, *p* < 0.001). There was no overlap in the dyslexia scores between the two groups. In keeping with the checklist guidelines [149], the control participants of the present study scored 41 or below while the dyslexic participants scored 45 or above.

No trained or professional musicians participated in the present study, though the musicality scores of the sample varied greatly, from 0 (no musical training received) to 21 (a relatively high level of musical experience). The majority of the participants had some music training: ten (out of thirteen) dyslexic participants and seven (out of eleven) controls reported to have learned an instrument or taken singing classes. Overall, the dyslexic participants of the study (mean: 7.4, SD: 6.4) engaged with music training more than the participants of the non-dyslexic control group (mean: 4.0, SD: 3.7). However, there was a lot of individual variability and the difference between the two groups of participants was not significant (t(19.8) = 1.57^3^, *p* = 0.13).

A series of post-hoc power analyses were run in Rstudio (R-version 4.1.0), using the *simr* package [151]. The power to detect group differences in five planned mixed-effects models was estimated, given the sample size. Based on 10,000 simulations of the best-fit models, the power to detect a group difference of 50 ms response time at α = 0.05 was above 60% in both tasks, the phoneme monitoring (72.15%) and the SMS (absolute asynchrony: 65.80%; signed asynchrony: 100%). Above 90% was the power to detect a group-level difference at α = 0.05 in the logistic mixed-effects models comparing the hit/miss responses during the phoneme monitoring (99.92%), and the likelihood of the synchronization with syllable nuclei during SMS (100.0%).

### 2.6. Data Preparation

The timing of raw tapping data collected by PsychoPy was first corrected by subtracting the delay of the Arduino microcontroller device (1 ms). Subsequently, the individual taps of each participant were aggregated across eight repetitions of each experimental sentence. The aggregated data were then subject to a Gaussian kernel density estimation, performed using the *ggplot2* library in R [152]. This procedure created a smoothed representation of tapping distributions for each participant and sentence. The resulting density displays are reflective of the participants’ preference to synchronize with specific anchor points in the sentence [50]. A higher density peak at a certain timepoint within a sentence indicated that this timepoint consistently attracted taps throughout sentence repetitions, thus highlighting the foci of rhythmic attention during speech perception [59,153]. Examples of the density distributions describing the SMS performance of the dyslexic vs. the control group recorded in the sentence “The couple watched the stars twinkle” are given in Figure 1. Subsequently, the findpeaks-function from the R-package *pracma* [154] was applied to each sentence, in order to identify the SMS peaks with a 40% threshold of the maximal peak value that occurred within a 100 ms distance from each other. Given that vowel onsets have been shown to constitute stable SMS anchors in different types of verbal stimuli [48,49,50] (Lin & Rathcke, 2020; Rathcke et al., 2019, 2021), the present study analyzed SMS with manually defined nucleus onsets, by calculating distances between the identified SMS peaks and the nearby (±120 ms) nucleus onsets [50].

Temporal distances between the temporal locations of tapping peaks and the nearby onsets of syllable nuclei were calculated as an index of participants’ SMS performance. Absolute asynchronies represented the synchronization accuracy with syllable nuclei while signed asynchronies reflected the tendency to anticipate, or to lag behind, a nucleus. Onsets of syllable nuclei were manually annotated by an expert phonetician (the first author). The presence of voicing, formant structure and the time course of signal intensity guided the segmentation. The acoustic and auditory criteria were combined in the segmentation of vowel onsets in post-sonorant contexts. Similarly, asynchronies between the vowel onset in [bi:] and the nearby tap were calculated in the paced tapping tasks. Finally, to characterize participants’ performance in the unpaced tapping task, the mean duration of inter-tap-intervals (ITIs) and the coefficient of their variability (CV, calculated as SD(ITI)/mean(ITI)) were measured for each participant in the spontaneous and fast task condition. 

### 2.7. Data Analyses

All analyses reported below were conducted in Rstudio running R-version 4.1.0., using the packages *lme4* [155], *emmeans* [156], *ggplot2* [152], and *sjPlot* [157].

## 3. Results

### 3.1. Group Differences in Spontaneous Movement and Time-Keeping

Neither the two unpaced tapping tasks (measuring the movement ability at a fast or a comfortable rate) nor the two paced tapping tasks (measuring the synchronization accuracy with targets presented at the rate of 300 or 600 ms) showed any significant group-level differences. Both participant groups had a non-clinical profile of their general SMS abilities [57], as shown in Table 3.

### 3.2. The Phoneme Monitoring Task

A logistic mixed regression (estimated using ML and the Nelder-Mead optimizer) was fit to hit/miss responses collected during the phoneme monitoring task. A linear mixed regression analysis was performed on the reaction time data (log-transformed, [158]). In both analyses, predictors included individual SMS rates collected in the SMS task and participants’ music training (both scaled and centered around the mean), rise-time duration of the consonant-to-vowel transition (on a logarithmic scale), relative intensity of the syllable (scaled and centered around the mean), syllable weight (strong/weak), target type (sonorant/obstruent), target position in the syllable structure (onset/coda) and the target being part of a consonant cluster or not (1/0). An interaction of these predictors with *group* (dyslexic/control) was tested in both models. *P**articipant* and *sentence* were fitted as crossed random effects. Maximal random effect structure was retained if the models converged and did not produce a singular fit [159]. The total explanatory power of the best-fit models was quite high (conditional R^2^ of the logistic model = 0.27, that of the linear model = 0.48), with a smaller contribution of the fixed effects alone (marginal R^2^ of the logistic model = 0.10, that of the linear model = 0.04). 

Despite large individual variability in the data, no predictor entered an interaction with *group*. The two groups of participants did not differ in their accuracy or reaction times when performing the phoneme monitoring task, on either class of the target phonemes; therefore, individual sensitivity d’ included responses to all targets. There was no differential influence of the three acoustic parameters (rise-time, duration, intensity). Individual SMS rates or music training did not affect the group performance significantly. These results are summarized in Table 4 and Table 5. 

### 3.3. SMS with Linguistic Stimuli

#### 3.3.1. SMS Likelihood

A logistic mixed-effects regression was performed to test for the likelihood of a tapping peak being present (1) or absent (0) in the proximity of a nucleus onset (±120 ms around the onset, [50]). Predictors included the metrical weight of the syllable (strong/weak), the rise-time of the amplitude envelope around the nucleus onset, the nucleus duration (both on a logarithmic scale), nucleus intensity (scaled and centered around the mean), participants’ sensitivity to acoustic cues of phonemes (d’) and their musicality scores (both individual measurements were scaled and centered around the group mean). Two-way interactions of these predictors with *group* (dyslexic/control) were tested and removed if an interaction did not help to improve the model fit or caused model convergence issues. *Participant* and *sentence* were fitted as random effects. We started with a maximal random effect structure and retained those random effects that allowed the models to converge (Barr et al., 2013). We changed the default optimizer (to “bobyqa”) and increased the number of iterations from default 10,000 to 100,000, to combat model convergence issues. The total explanatory power of the best-fit model was substantial (conditional R^2^ = 0.28), with a relatively large contribution of the fixed effects alone (marginal R^2^ = 0.20). Pairwise comparisons of the group performance report estimated marginal means (or least-squares means), obtained using the *emmeans* package in R [156].

The relevant results are summarized in Table 6. The best-fit model contained three two-way interactions. The interaction of *group* and *metrical weight* indicated that dyslexic participants were less likely to synchronize with metrically weak syllables than non-dyslexic controls (β = 0.61, SE = 0.26, z = 2.36, *p* < 0.05) but did not differ in their SMS with metrically strong syllables (β = 0.08, SE = 0.32, z = 0.24, *p* = 0.81, see Figure 2A). Individual variability in d’ also interacted with *group*. While individual sensitivity to acoustic cues of a phoneme in a nonsense string (d’) did not affect the SMS in dyslexic participants (β = 0.19, SE = 0.15, z = 1.23, *p* = 0.22), such individual sensitivity of the control participants was predictive of their SMS. Participants who were better able to correctly segment and identify phonemes in nonsense strings also synchronized with more syllable nuclei (β = 0.50, SE = 0.21, z = 2.44, *p* < 0.05, see Figure 2B). Moreover, an interaction of individual musicality scores with *group* showed that higher levels of musical training led to a higher SMS likelihood in the controls (β = 0.72, SE = 0.26, z = 2.77, *p* < 0.01) but to a lower SMS likelihood in dyslexic participants (β = −0.38, SE = 0.15, z = −2.50, *p* < 0.05, Figure 2C).

An additional logistic mixed-effects regression was fit to a subset of the data containing SMS with weak syllables only. The structure of the model mirrored the above, with the only replacement of the factor *metrical weight* by the factor *nucleus type* (vowel/sonorant). The best-fit model is given in Table 7. The model’s explanatory power was moderate (conditional R^2^ = 0.19) with a smaller contribution of the fixed effects alone (marginal R^2^ = 0.10). The corresponding effects are plotted in Figure 3. Accordingly, both groups were equally more likely to synchronize with vocalic than sonorant nuclei (β = 0.42, SE = 0.13, z = 3.29, *p* < 0.01, Figure 3A) while they differed with regard to the individual effects of musical training (Figure 3C) and sensitivity (d’, Figure 3B) on their SMS. More specifically, similar interactions of *group* with the individual abilities were found in the subset of data as well as in the whole dataset (cf. Figure 2B,C).

#### 3.3.2. Absolute Asynchrony

We fitted a linear mixed model (estimated using REML and “nloptwrap” optimizer) to predict logarithmically transformed absolute asynchronies with metrical weight of the syllable (strong/weak), nucleus duration, and rise-time of the amplitude envelope around the nucleus onset (both on a logarithmic scale), nucleus intensity (scaled and centered around the mean), participants’ sensitivity (d’), musicality scores and their individual absolute asynchronies measured in the paced tapping tasks (all individual values were scaled and centered around the group mean). *Participant* and *sentence* were defined as random effects. We tested for two-way interactions of each predictor with *group* (dyslexic/control) and included a maximal random effect structure, successively reducing the complexity of the initial model by removing predictors that were not significant and random effects that caused convergence or singular fit issues. The total explanatory power of the best-fit model was rather weak (conditional R^2^ = 0.02), equaling to the total contribution of the fixed effects alone (marginal R^2^ = 0.02). 

The magnitude of the effects from the best-fit model is plotted in Figure 4. Accordingly, all participants (regardless of the group) showed longer asynchronies with metrically weak than with metrically strong syllables (β = 0.23, SE = 0.05, t = 5.03, *p* < 0.001, Figure 4A). A differential effect of the group membership was again found for participants with variable sensitivity (d’): while control participants with higher d’-scores tended to show smaller asynchronies (β = −0.06, SE = 0.03, z = −2.06, *p* = 0.058), dyslexic participants with higher d’-scores had larger asynchronies in their SMS with vowel onsets (β = 0.13, SE = 0.04, z = 3.08, *p* < 0.01, Figure 4B). In other words, the more phonemes the control participants were able to spot, the more accurate they were synchronizing with the nucleus onsets, while the opposite is true for the dyslexic participants: the more phonemes they were able to correctly identify in the phoneme monitoring task, the less accurate they tapped with nucleus onsets. A summary of the relevant results is given in Table 8.

#### 3.3.3. Signed Asynchrony

A linear mixed model (estimated using REML and the “nloptwrap” optimizer) was fitted to signed asynchrony (scaled and centered around the group mean). Predictors were the metrical weight of the syllable (strong/weak), logarithmically transformed nucleus duration and rise-time of the amplitude envelope, vowel intensity (scaled and centered around the mean), participants’ musicality scores, their sensitivity d’ obtained in the phoneme monitoring task and their signed asynchronies collected in the paced tapping tasks (all individual values were scaled and centered around the group mean). Two-way interactions of each predictor with *group* (dyslexic/control) were tested. Random effects were *participant* and *sentence*. The first model included a maximal random effect structure. The final model retained only those random effects that produced no convergence or singular fit issues. The total explanatory power of the best-fit model was weak (conditional R^2^ = 0.03), with a smaller contribution of the fixed effects alone (marginal R^2^ = 0.02). 

The estimated effects of the best-fit model are shown in Figure 5. Accordingly, all participants (regardless of their group affiliation) tended to predict—that is, tap ahead of—longer syllable nuclei (β = −13.71, SE = 1.70, t = −8.39, *p* < 0.001, Figure 5A). A differential effect of the group came to the fore in participants with variable signed asynchronies measured in the paced tapping task (Figure 5B). More specifically, signed asynchronies produced in the task with simple linguistic stimuli playing repetitions of the syllable [bi:] at 600 ms IOI could predict signed asynchronies with the natural linguistic stimuli, but only in the control (β = 6.01, SE = 1.68, t = 3.57, *p* < 0.01) and not in the dyslexic (β = 0.91, SE = 1.39, t = 0.66, *p* = 0.52) group of participants. There was a positive linear relationship in the neurotypical participants’ performance, i.e., those predicting a vowel onset in simple, isochronous stimuli, also predicted nucleus onsets in complex, natural stimuli. The relevant results are summarized in Table 9.

## 4. Discussion

The present study set out to investigate rhythm perception and entrainment in dyslexia, and aimed at making a contribution toward a deeper understanding of prosodic deficits and the underlying causes of dyslexia by studying adult dyslexic adults’ performance in an SMS task with natural speech. The following section discusses the results of the study with respect to the originally outlined open questions and hypotheses.

### 4.1. Rhythm Perception and Motor Entrainment in Dyslexia

Rhythm perception and motor entrainment were studied by means of a previously developed SMS paradigm that utilized finger tapping to the subjectively perceived beat of looped sentences [50]. To our knowledge, the present study is the first to attest sensorimotor synchronization with natural, syntactically complex and metrically irregular sentences in dyslexic adults. Even though our materials lacked regular occurrences of strong and weak syllables, and contained syllables with non-vocalic nuclei (see Appendix A), they did not generally cause more difficulties to the dyslexic participants than to the age-matched controls during SMS. The performance observed in the dyslexic adults’ SMS was not indicative of a prevalent issue with the syllabic entrainment *per se*, as Leong and Goswami [22] suggested. Instead, we found that the dyslexic participants of the study synchronized with the nuclei of metrically weak syllables less frequently than the controls, without showing any differences in the synchronization accuracy, or prediction tendency for these syllables. At the same time, both groups generally produced fewer and less precise taps with weak syllable nuclei, particularly if those were occupied by sonorants. Both groups synchronized more frequently and more accurately with onsets of strong syllable nuclei. The latter finding is in line with the results discussed by Leong and Goswami [22], suggesting that adults with developmental dyslexia have no entrainment deficits at larger timescales such as inter-stress intervals.

Given the similarity of the overall performance across the two groups, these findings cannot be easily reconciled with the idea that the hierarchical relationships between stressed and unstressed syllables are inadequately encoded in dyslexic listeners (e.g., [22,110,111,115]). Overall, the metrical weight of syllables was equally well reflected in all participants’ SMS, with the only key difference that weak syllables were significantly less at the center of rhythmic attention in the dyslexic participants than in the neurotypical controls (cf. [153]). As far as the ongoing discussions of neural entrainment deficits in dyslexia are concerned, the results of the present study can be reconciled with the growing body of research reporting deviant patterns in the delta range (~0.5–4 Hz, [160,161]) which cover the timescale of plausible intervals spanning the distance between unstressed units of the prosodic hierarchy.

The reduced attention to weak syllables can hardly arise as a purely psychoacoustic effect of locally varied rise-time, duration or intensity, given that these measures had little influence on the participants’ performance in the SMS task. Even though the rise-time of the experimental materials measured significant differences between strong vs. weak syllables and between weak syllables with vocalic vs. sonorant nuclei, such variability in signal acoustics did not have any differential effects on the participants from the two experimental groups. This finding suggests that rhythm processing in natural speech may not be influenced by signal rise-time as is sometimes assumed [22,127,128,129], and that it plays little role in the dyslexic perception of speech rhythm as previously discussed in light of correlational evidence (e.g., [110,115,125,126]). Importantly, the results of the present study stem from an experimental design, and provide innovative evidence in the extension of previous, exclusively correlational studies that showed some inconsistencies of the rise-time effect in dyslexia observed across different tasks (e.g., [110,162]).

One acoustic-prosodic parameter, if any, might have indirectly shaped the group difference in SMS with weak syllables found in the present study. Since the relative intensity differed systematically between strong vs. weak syllables (but not between the two types of weak syllables), the reduced sensorimotor synchronization with weak syllables observed among dyslexic participants could be related to their rhythmic attention being more extensively reliant upon perceptual prominence than the rhythmic attention of the neurotypical participants. Acoustic intensity, corresponding to the psychoacoustic impression of loudness, is known to dominate the perception of auditory prominence, at least in English [121], and is the only acoustic parameter of the materials that displays a (distant) similarity with the group results found in the SMS experiment. The role of intensity in dyslexic speech processing would thus benefit from a further experimental replication with new speech materials exhibiting prominence variability on a larger scale, and including emphatic and contrastive prominence.

A further group-level finding of the SMS experiment demonstrates that the tendency to anticipate upcoming rhythmic events differed between the dyslexic and the neurotypical groups. In contrast to previous research by Thomson and Goswami [126], who found more anticipation among their dyslexic participants synchronizing with a metronome, there was no general anticipation tendency observed in the present group of dyslexic adults synchronizing either with simple one-word loops or with complex sentence loops. Rather, our results concerning signed asynchronies indicate a difference in the group-level consistency of SMS behaviors. Among the control participants of the study, there was a high level of consistency in their SMS performance across rhythmically simple and complex tasks. The control participants’ synchronization with natural speech mirrored their performance with simple, isochronous sequences containing a looped monosyllabic word presented at 600 ms IOI. This high level of within-group consistency stands in a stark contrast to the results of the dyslexic participants whose individual tendency to predict an upcoming rhythmic event did not transfer from a simpler to a more complex rhythmic task. As noted in many previous studies, individuals diagnosed with developmental dyslexia frequently form no homogenous group and show high interpersonal variability in their behaviors [163]. Different “routes to failure” [132] might be responsible for the diverse behavioral profiles observed in individuals diagnosed with developmental dyslexia, including their non-clinical performance in certain tasks [164,165,166,167].

### 4.2. Phonological Awareness and Prosody

The findings of the present study contribute to the understanding of the role of prosody in phonological awareness. We considered several possibilities that could explain how the ability to perceive and encode rhythmic structure in speech might be related to the segmental phonological skills, and tested these relations by studying the cross-over effects of the participants’ performance in the two tasks: (1) phoneme monitoring that required segmentation and the processing of variable acoustic cues, and their mapping to the abstract representations of sonorant and obstruent phonemes; and (2) SMS that relied on the perception of a rhythmic beat and motor entrainment to the perceived beat in time. As far as phoneme monitoring is concerned, our prediction that dyslexic listeners would show more difficulties in identifying obstruents rather than sonorants (cf. [35,142]) was not borne out by the data, and there was also no effect of the participants’ SMS rates on their identification of phonemes embedded in nonsense strings. However, participants’ sensitivity d’ derived from their performance in the phoneme monitoring task explained some individual and group-level variability found in SMS.

Two aspects of participants’ SMS—frequency and accuracy (but not anticipation)—could be, at least partly, predicted by their performance during phoneme monitoring. Among the control participants, the individual ability to correctly segment and identify phonemes in nonsense strings was predictive of their tendency to attend to the rhythmic beat of each and every syllable. The more phonemic targets were correctly identified, the more syllabic nuclei were also synchronized to. This effect was absent in the dyslexic participants’ data, possibly due to a high level of individual variability observed across the two tasks (cf. [163]). However, participants with dyslexia differed significantly from the control group in their synchronization accuracy, and showed opposite cross-over effects with regards to the sensitivity d’. While neurotypical participants tapped more accurately in time with the sentence beats if their sensitivity d’ in the phoneme monitoring task was high, dyslexic participants with the highest d’ produced the least accurate asynchronies in the SMS task. 

Overall, individual performance displayed a high level of cross-task congruence in the neurotypical but not in the dyslexic group of participants (cf. [163]). In a way, dyslexic results can be viewed such that an increased accuracy in a segmental phonology task leads to a delayed response time in a rhythmic synchronization task. This might be reflective of individually variable perception strategies fueled by specific speed–accuracy trade-offs in the context of limited cognitive resources [168], and adds to the previous research documenting auditory short-term memory issues in dyslexia (e.g., [169]; see [29], for an overview). Upon review of a large body of experimental findings, Ramus and Szenkovits [29] suggest that phonological representations are likely to be intact in individuals diagnosed with developmental dyslexia, but their short-term memory often fails to retrieve them when necessary. More recently, Tierney et al. [170] demonstrated a link between reading ability and memory for rhythm. Our results support previous observations, and add further insights from individual variability among adults with dyslexia. Those dyslexic participants of our study who established better access to their segmental phonology, seem to have traded in the response time speed in a rhythmic-prosodic task.

Taken together, these findings suggest that previously documented deficits in the perception and encoding of rhythmic structures in speech [22,43,110,111,112,113,114,115] are unlikely to constitute the primary issue in dyslexia, but might rather be moderated by an individual severity of difficulties in the access to segmental phonological representations, arising as a consequence of a limited ability to attend to the relevant acoustic cues to segmental phonological contrasts. Dyslexic deficits in phonological awareness may therefore be caused by insensitivity to sub-lexical units and consequently to acoustic cues encoding higher-level structures such as syllable boundaries, lexical stress and metrical hierarchies. Importantly, the present study outlines an innovative approach for understanding the sources of individual variability in the frequently heterogenous dyslexic group performance [164,165,166,167]. Given that no significant group effects were observed during phoneme monitoring despite a high level of individual variability, this approach might in the future help to uncover hidden relationships between dyslexic participants’ non-clinical performance in certain tasks and their underlying deficits.

### 4.3. Lack of an Acoustic Influence on the Phonemic and Prosodic Processing

The present study focused on the acoustic parameters such as rise-time, duration and intensity, and addressed their role in dyslexic processing of natural and nonsense speech. Instead of taking up the prevalent approach from previous research and employing purely psychoacoustic tests to estimate individual sensitivity to these aspects of acoustic signals (e.g., [83,110,115,129,130,131]), this study estimated the impact of variable signal rise-time, duration and intensity on the dyslexic and control participants’ performance in two experimental tasks, phoneme monitoring and SMS. If an auditory processing deficit of (any of) the above-mentioned acoustic parameters was the main cause underlying dyslexic difficulties as sometimes suggested (see [117], for an overview), we would expect to find an effect of (at least some of) these parameters on the dyslexic participants’ performance in the verbal tasks. However, the results showed that hardly any of these parameters accounted for the variance either in the phoneme monitoring task, or in the SMS task. Only the duration of a syllable nucleus had an effect on SMS and was predictive of the amount of anticipation. Longer syllable nuclei attracted larger signed asynchronies, i.e., were more anticipated, than shorter nuclei. No further effects of the acoustic properties of syllables were found. Moreover, the dyslexic and the neurotypical adult participants of the present study did not differ in this regard. 

These results speak to the proposal put forward by Protopapas [132] that the links between auditory difficulties observed in dyslexic participants’ non-verbal and verbal processing (e.g., [83,110,115,129,130,131,163]) might not be causational as is sometimes suggested, but concomitant, deriving from the same (multitude of) factors and affected by the same (or neighboring) cortical structures. Experimental research with non-correlational designs is needed to further elaborate on the nature of the relation between verbal and non-verbal auditory deficits in individuals of different ages diagnosed with developmental dyslexia. Even though the idea that an auditory processing deficit would cause inaccurate speech perception and therefore lead to inadequate phonological representations and subsequent reading difficulties appeals for its simplicity, conflicting evidence among existing studies (e.g., [171,172,173]; see [166] for a review) also points toward the need to revise the hypothesis of a general auditory origin of dyslexia.

### 4.4. On the Role of Music Training in Dyslexia

The role of music training in speech processing has been discussed in a number of studies, demonstrating benefits for aging adults, difficult listening conditions and short-term memory capacity (e.g., [61,62,63,64,65,66,67,69,70,174]). Fueled by the auditory processing hypothesis of dyslexic deficits, music-based therapies have been developed and implemented, though with mixed results, as a means of treating dyslexia in children (e.g., [80,81,82,83]). Assuming that music practice enhances general auditory skills and thus benefits speech and language processing (e.g., [60,85,86,87]), we hypothesized that individually variable levels of previous music training would influence participants’ performance in the two experimental tasks of the present study—phoneme monitoring and SMS—and expected the effect of music training to potentially outweigh the dyslexic status of participants (cf. [88]).

Unexpectedly, previous music training did not play a significant role in both experimental tasks. Only the rhythmic-prosodic SMS task of the present study showed the hypothesized effect, with notable differences between the two groups of participants. This result concerned the likelihood, but not the asynchrony, of rhythmic synchronization with the syllabic targets. In the neurotypical group, we observed an increase in the tendency to synchronize with each syllable nucleus in participants with higher levels of music training which is in line with the predictions made by the general auditory skills account (e.g., [60,85,86,87]).

In contrast, dyslexic participants with more music training synchronized to fewer nuclei. Given that the main difference between the two groups lies in their synchronization with weak syllables, the effect may be interpreted as a shift of rhythmic attention to exclusively strong syllables in dyslexic participants with a high level of musical training. Such increased selective sensitivity to stressed syllables might be a compensatory mechanism that dyslexic listeners acquire while learning how to manage the complexities of speech and language processing in light of their limited cognitive resources. Discussing “the enigma of dyslexic musicians”, Weiss et al. [169] concluded that deficits of the auditory short-term memory capacity remain the key issue even in very skilled musicians with developmental dyslexia, regardless of their extensive auditory training during music practice. Since the perception of lexically stressed syllables is of the utmost importance for speech segmentation and lexical access [106,107], a decrease of rhythmic attention paid to unstressed syllables would save cognitive resources while preserving the key functionality in the perceptual language system. Using a prosodic skills battery [175], previous research demonstrated that the ability of dyslexic children to use linguistic prosody for communicatively relevant purposes such as semantic or syntactic disambiguation was commensurate with their general linguistic abilities [176]. This means that some prosodic processing difficulties in dyslexia can indeed be well compensated in communicative interaction, even in young children with developmental dyslexia.

Unlike its high relevance in the SMS task, music training did not play an important role in the phoneme monitoring task (either alone or in an interaction with the participant group). This finding suggests that the benefits of music training may be specific to the perception of speech prosody and do not generalize to the perception of segmental phonology. Even though word-level segmentation skills improve in listeners with a higher level of musical skills [67,68], the auditory benefits do not seem to transfer to the phoneme-level segmentation that was involved in the phoneme monitoring task of the present study. Since language and music share primarily prosodic-acoustic features related to pitch, timing and timbre [86], music training is likely to bring about an exclusive advantage for the processing of these aspects of auditory signals [177,178]. The exact mechanisms underpinning phonological awareness and the role of musical and prosodic skills in its development are yet to be fully established and require further research.

## 5. Conclusions

The present study focused on dyslexic adults with and without music training, and investigated their rhythm perception in, and entrainment to, natural speech [50], along with their ability to perceive segmental phonological contrasts. Our results showed that both dyslexic and neurotypical participants attended less to metrically weak than to metrically strong syllables, particularly if the syllabic nuclei were occupied by sonorants; however, participants with dyslexia did so to a greater extent than the neurotypical, age-matched controls. Overall, participants’ performance during phoneme monitoring was predictive of their performance during SMS, but not vice versa, suggesting that prosodic processing is not the primary deficit in dyslexia as previously suggested [22,43,110,111,112,113,114,115]. 

Against expectation, signal acoustics such as rise-time, duration and intensity played no role in the group performance during either the segmental or the prosodic task, calling for a revision of the general auditory deficit hypothesis of dyslexia put forward on the basis of correlational evidence (e.g., [127,128,129]). This conclusion is further corroborated by the finding that higher levels of music training did not lead to an improved performance of our dyslexic participants across all tasks as the general auditory deficit hypothesis would predict. Rather, the effect of increased music training was specific to the rhythmic task, and its direction was suggestive of a compensatory strategy in musically trained adults with dyslexia as a means of coping with limited short-term memory resources (cf. [29,166]). In their review of the controversial findings surrounding the nature of dyslexic deficits, Ramus and Ahissar [166] concluded that prosody perception and/or production have only been found impaired in the tasks that involved metalinguistic judgements of prosody or other difficulty factors such as the presence of background noise, an increased short-term memory load or time pressure in a task at hand. The present results complement and corroborate this proposal by contributing innovative evidence to the understanding of individual pathways to compensation for language processing issues among dyslexic adults (cf. [132]). The study had not been designed to investigate the role of short-term memory in dyslexic processing of prosody, and its conclusions concerning the involvement of the cognitive resources require a follow-up investigation involving measurements of auditory short-term memory capacity, in order to corroborate the present findings.

Of course, the results of the present study reflect, and are limited to, the performance of well-compensated adults who had been diagnosed with developmental dyslexia as children and reported merely mild dyslexic symptoms at the time of testing. It is possible that dyslexic children will show different experimental profiles indicative of more substantial deficits, due to the lack of compensation strategies that adults may have developed over time (e.g., [179]). A longitudinal or cross-sectional study would be beneficial in establishing individual trajectories in dyslexic language processing, and the role of music training in this development. 

The conclusions above support the cognitive theories of dyslexia (cf. [165,180]), at least as far as auditory processing is concerned. Independent evidence exists on poor readers’ deficits in senses other than audition, including vision (e.g., [181,182]), somatosensation [183], proprioception [184] and audiovisual integration ([185]; see [186], for a review). In order to advance the understanding of the underlying causes of developmental dyslexia, future research will benefit from cross-disciplinary approaches and comprehensive, non-correlational designs to study the complexities involved in reading ability and skills.

## Figures and Tables

**Figure 1 brainsci-11-01303-f001:**
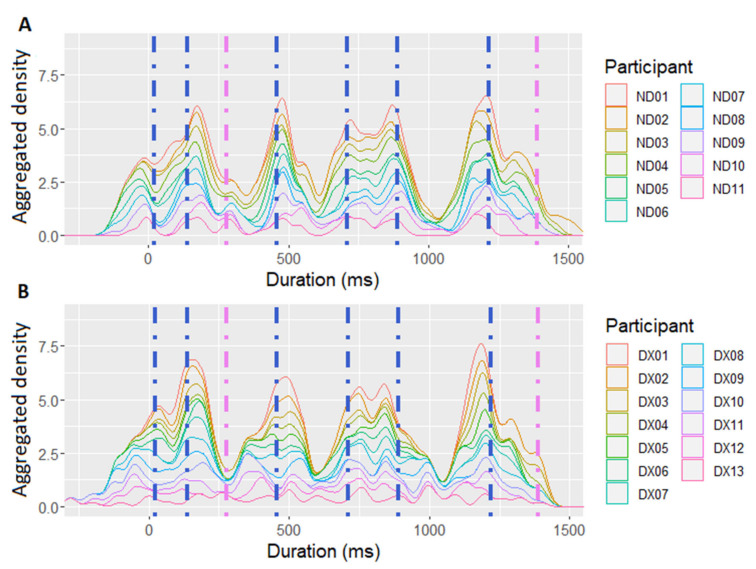
Aggregated densities of SMS with the test sentence “The couple watched the stars twinkle”. (**A**) displays the performance of the control group, and (**B**) that of the dyslexic group. Vertical blue lines indicate onsets of vowels, and pink lines indicate onsets of sonorant nuclei.

**Figure 2 brainsci-11-01303-f002:**
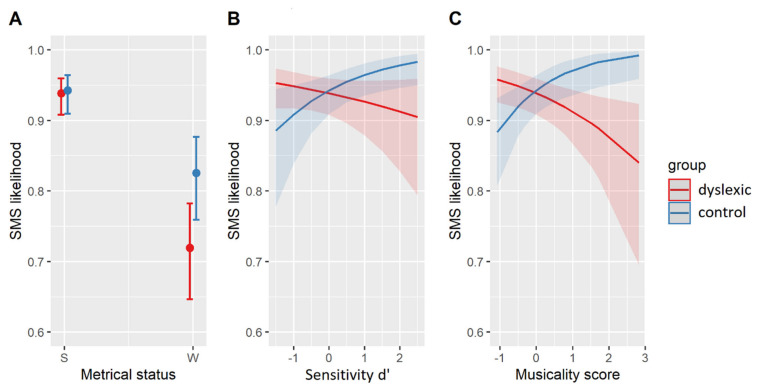
Likelihood of SMS estimated in adult participants with dyslexia (red lines) vs. controls (blue lines), depending on (**A**) metrical weight of a syllable, (**B**) individual sensitivity to acoustic cues d’ and (**C**) individual musicality scores.

**Figure 3 brainsci-11-01303-f003:**
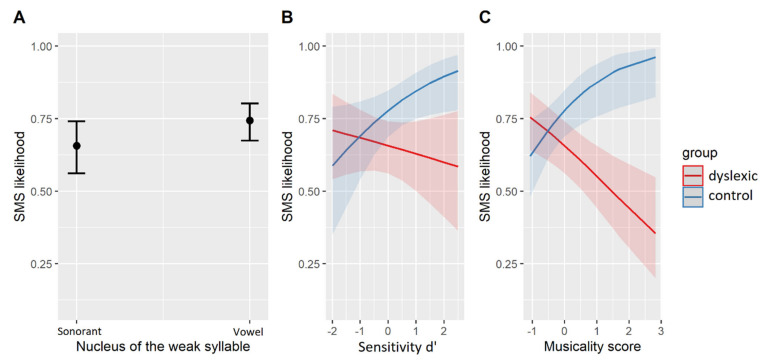
Likelihood of SMS estimated in weak syllables, for (**A**) the main effect of syllable nucleus, (**B**) the interactions of group and d’-scores and (**C**) the interaction of group and musicality scores. Dyslexic data are plotted in red, control data in blue.

**Figure 4 brainsci-11-01303-f004:**
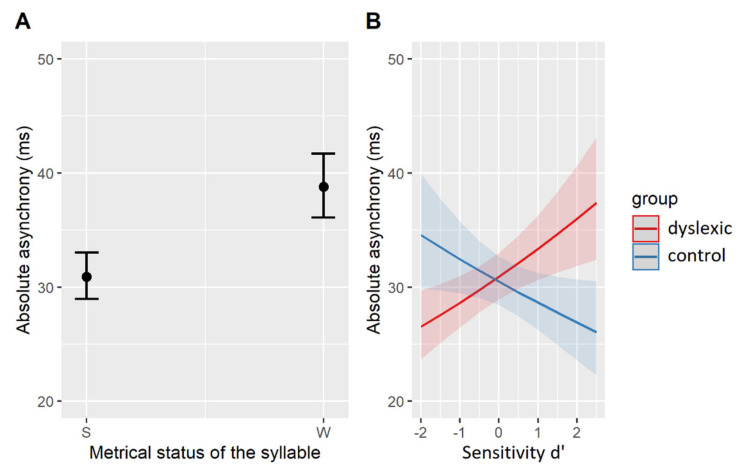
Predicted values of absolute asynchronies in (**A**) strong vs. weak syllables, and (**B**) dyslexic (red) vs. control (blue) participants with variable d’-scores (plotted on the *x*-axis).

**Figure 5 brainsci-11-01303-f005:**
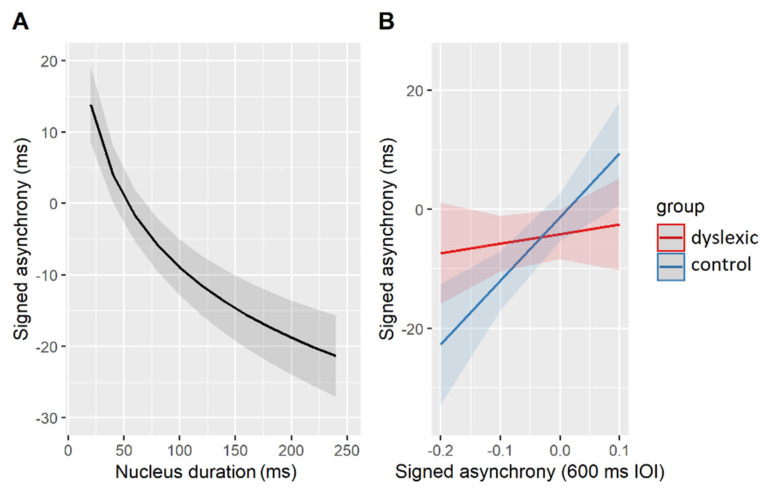
Predicted signed asynchronies in (**A**) syllable nuclei of variable duration, and (**B**) dyslexic (red) vs. control (blue) participants with variable signed asynchronies measured in the paced tapping task with the targets presented at IOI of 600 ms (plotted along the *x*-axis). On the *y*-axis, 0 ms refers to the nucleus onset.

**Table 1 brainsci-11-01303-t001:** Means (and standard deviations) of the acoustic measurements taken from the experimental stimuli, along with the corresponding statistical results.

Acoustic Factors	Obstruents /p/, /k/	Sonorants /m/, /l/	ANOVA	
			F	*p*
*Rise-time*	66.21 (26.69)	69.28 (42.95)	F(1, 38) = 0.07	0.79
*Duration*	50.17 (17.71)	87.95 (22.71)	F(1, 38) = 32.76	<0.001
*Relative intensity*	0.89 (0.04)	1.00 (0.05)	F(1, 38) = 62.21	<0.001

**Table 2 brainsci-11-01303-t002:** Means (and standard deviations) of the acoustic measurements taken from the experimental stimuli, along with the corresponding statistical results.

Acoustic Factors	S	W	ANOVA		W (Vowel)	W (Sonorant)	ANOVA	
			F	*p*			F	*p*
*Rise-time*	50.41 (18.58)	37.68 (15.47)	F(1, 194) = 26.93	<0.001	34.14 (14.15)	50.47 (13.28)	F(1, 118) = 27.84	<0.001
*Duration*	92.98 (43.48)	60.82 (33.02)	F(1, 194) = 34.38	<0.001	51.53 (28.96)	94.39 (23.91)	F(1, 118) = 47.82	<0.001
*Relative intensity*	1.09 (0.05)	1.04 (0.05)	F(1, 192) ^1^ = 35.88	<0.001	1.04 (0.05)	1.06 (0.05)	F(1, 116) ^1^ = 1.35	0.25

^1^ The intensity measurements could not be taken for two syllables out of a total of 196 syllables tested in the present study (see Appendix A).

**Table 3 brainsci-11-01303-t003:** Summary of group means (and standard deviations) measured in spontaneous and paced motor tasks, along with the corresponding statistical results.

Measurements Taken		Dyslexic Group	Control Group	Welch’s Two Sample *t*-Test		Cohen’s d
				t	*p*	
Spontaneous tapping at a comfortable rate	ITI (ms)	498 (152)	427 (151)	t (21.39) = 1.14	0.27	0.47
	CV	0.15 (0.16)	0.07 (0.03)	t(13.02) = 1.74	0.10	0.69
Spontaneous tapping at a fast rate	ITI (ms)	185 (28)	195 (52)	t(14.74) = 0.56	0.58	−0.24
	CV	0.15 (0.13)	0.22 (0.24)	t(14.95) = 0.90	0.38	−0.38
Paced tapping at 300 ms IOI	ITI (ms)	294 (22)	291 (26)	t(19.42) = 0.28	0.78	0.12
	CV	0.09 (0.07)	0.08 (0.04)	t(20.85) = 0.29	0.77	0.12
	Signed asynchrony (ms)	3 (19)	−11 (20)	t(20.96)= 1.76	0.09	0.72
	Signed asynchrony (proportion IOI)	0.01 (0.06)	−0.04 (0.07)	t(20.96)= 1.76	0.09	0.72
	Absolute asynchrony (ms)	32 (24)	33 (20)	t(21.99) = 0.04	0.97	−0.02
	Absolute asynchrony (proportion IOI)	0.11 (0.08)	0.11 (0.07)	t(21.99) = 0.04	0.97	−0.02
Paced tapping at 600 ms IOI	ITI	687 (327)	593 (19)	t(12.10) = 1.03	0.32	0.41
	CV	0.13 (0.24)	0.07 (0.06)	t(13.85) = 0.89	0.39	0.35
	Signed asynchrony (ms)	−25 (37)	−23 (34)	t(21.81) = 0.15	0.88	−0.06
	Signed asynchrony (proportion IOI)	−0.04 (0.06)	−0.04 (0.06)	t(21.81) = 0.15	0.88	−0.06
	Absolute asynchrony (ms)	49 (20)	45 (23)	t(20.00) = 0.40	0.69	0.17
	Absolute asynchrony (proportion IOI)	0.08 (0.03)	0.07 (0.03)	t(20.00) = 0.40	0.69	0.17

**Table 4 brainsci-11-01303-t004:** Summary of the logistic mixed-effects model best fitting the accuracy data.

Factor	AIC	df	𝝌^2^	*p*
*Cluster*	1110.0	1	10.73	<0.01
*Target type*	1109.7	1	6.94	<0.01
*Syllable position*	1110.5	1	7.71	<0.01
*Relative intensity*	1116.4	1	13.66	<0.001
*Group*	1104.7	1	0.12	0.73
*Group × target type*	1106.6	1	0.58	0.44
*Group × rise-time*	1105.8	1	3.02	0.08
*Group × target duration*	1106.4	1	0.09	0.76
*Group × relative intensity*	1106.6	1	0.05	0.82
*Group × SMS rate*	1108.2	1	0.35	0.56
*Group × musicality*	1108.3	1	0.05	0.82

**Table 5 brainsci-11-01303-t005:** Summary of the linear mixed-effects model best fitting the RT data.

Factor	Sum Sq.	df	F	*p*
*Target type*	1.41	1	7.15	<0.05
*Relative intensity*	2.98	1	15.08	<0.001
*Group*	0.41	1	2.09	0.16
*Group × target type*	0.13	1	0.66	0.42
*Group × rise-time*	0.11	1	0.54	0.46
*Group × target duration*	0.09	1	0.48	0.49
*Group × relative intensity*	0.26	1	1.30	0.25
*Group × SMS rate*	0.19	1	0.97	0.33
*Group × musicality*	0.40	1	2.01	0.17

**Table 6 brainsci-11-01303-t006:** Summary of the logistic mixed-effects model best fitting the SMS probability data.

Factor	AIC	df	𝝌^2^	*p*
*Group × metrical weight*	4217.1	1	4.34	<0.05
*Group × individual d’*	4219.4	1	6.64	<0.01
*Group × musicality*	4223.6	1	10.79	<0.01
*Group × rise-time*	4216.2	1	0.86	0.35
*Group × nucleus duration*	4204.6	1	0.65	0.42
*Group × relative intensity*	4204.8	1	0.07	0.78

**Table 7 brainsci-11-01303-t007:** Summary of the logistic mixed-effects model best fitting the SMS probability data (weak syllables only).

Factor	AIC	df	𝝌^2^	*p*
*Nucleus type*	3186.6	1	8.31	<0.01
*Group × individual d’*	3182.4	1	4.17	<0.05
*Group × musicality*	3189.9	1	11.61	<0.001

**Table 8 brainsci-11-01303-t008:** Summary of the linear mixed-effects model best fitting the absolute asynchrony data.

Factor	Sum Sq.	df	F	*p*
*Metrical weight*	27.15	1	25.25	<0.001
*Group × metrical weight*	0.01	1	0.01	0.92
*Group × individual d’*	13.49	1	12.55	<0.01
*Group × musicality*	2.18	1	2.03	0.17
*Group × rise-time*	0.99	1	0.92	0.34
*Group × duration*	0.26	1	0.24	0.62
*Group × intensity*	1.29	1	1.20	0.27

**Table 9 brainsci-11-01303-t009:** Summary of the linear mixed-effects model best fitting the signed asynchrony data.

Factor	Sum Sq.	df	F	*p*
*Nucleus duration*	41,602	1	12.77	<0.01
*Group × metrical weight*	2645	1	0.82	0.36
*Group × individual d’*	741	1	0.23	0.64
*Group × musicality*	1614	1	0.50	0.49
*Group × signed asynchrony (600 ms IOI)*	29,176	1	8.92	<0.01
*Group × rise-time*	713	1	0.22	0.64
*Group × nucleus duration*	102	1	0.03	0.86
*Group × relative intensity*	10,639	1	3.30	0.07

## Data Availability

The data collected for the present study can be made available upon request.

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
