# Peer review of "Towards a Comprehensive Account of Rhythm Processing Issues in Developmental Dyslexia"

_brainsci, 2021, doi:10.3390/brainsci11101303_

Round 1

Reviewer 1 Report

The current paper describes a study designed to assess the relative contribution of rhythmic and phonemic deficits of dyslexics. Two groups of participants – 13 dyslexics and 11 controls – participated in the experiment. They completed a phoneme monitoring task, 4 tapping tasks, and one sensorimotor synchronization task where they tapped along with repeated sentences. Results demonstrate that participants’ performance on the phoneme monitoring task predicted their performance on the synchronization task, but not vice versa, leading the authors to conclude that the rhythmic deficits of dyslexics are likely due to challenges with phonetic perception.

The introduction does an excellent job summarizing what is known about the connection between rhythmic processing and dyslexia, I like the novelty of having participants tap along to non-isochronous sentences, and looking at weak as well as strong syllables. I also think it’s valuable that the authors looked at the influence of individual acoustic predictors (rise time, duration, intensity) on performance.

I have questions about the identification of the participants. The paper indicates that participants self-identified as dyslexic (or not), and that this designation was corroborated with a checklist. Without seeing the checklist, I wonder how sure the authors can be about the designation. It would be good to know if scores on the checklist have been previously shown to correlate with self-report, or other more objective assessments, of dyslexia. Moreover, was there any overlap in scores between the dyslexic and non-dyslexic participants?

I’m also concerned about power issues with these data. Are 13 and 11 in the two groups enough to demonstrate significant differences between groups? There’s the additional factor of musical training. My concern about low power is most important in thinking about the lack of differences in the phoneme monitoring results. Minimally, I think the authors need to provide a power analysis to show that they could uncover differences between the groups in the phoneme monitoring task. But I also worry that there is enough variability within the population of dyslexic adults (mentioned in the current paper, in fact) that this sample size cannot capture true differences. For example, Leong & Goswami (2014) tested almost twice this number of participants.

I think it would help for the authors to make explicit in the Method that the SMS task requires tapping in time with ALL syllables. I know this is pointed out in the Intro, but it would be helpful to be explicit in the Method as well.

Line 394: “tap on the tambourine” The tambourine needs to be introduced before this mention. It’s introduced in the next section on the SMS task.

Line 540: There’s an indication of a footnote, but I can’t find the footnote being referenced?

Line 552: Remind the reader what group is referring to here.

Line 589 – what is the test here (z = 2.36, p > .05) describing? Not clear, and should it be ‘p < .05’? There are other tests reported in this paragraph where the actual test isn’t clear. Please clarify.

I would recommend the authors cite Tierney, et al., 2021 (https://www.sciencedirect.com/science/article/pii/S0022096521001144), who also show a connection between reading ability and memory for rhythm.

Author Response

AUTHORS: We would like to thank the Editors and the Reviewers for the appreciation of our work, and for the critical engagement with our research that helped us improve the manuscript. There seems to have been a problem during typesetting of the manuscript. An earlier version of our submission was sent out for review. An updated version did not substantially differ from the earlier version in terms of content, only the abstract was amended and the structure re-shaped (sect. 1.3 and 1.4 were combined in sect. 1.2; sect. 1.2. moved to sect. 1.3 and renamed). This was done in response to the Main Editor’s suggestion, by way of strengthening the connection of the study with the theme of the special issue, and highlighting the previously documented motor issues in dyslexia. The revised version of the manuscript re-implements these previous changes.

We addressed all issues identified by the reviewer, and outlined the changes below.

R1-C1: I have questions about the identification of the participants. The paper indicates that participants self-identified as dyslexic (or not), and that this designation was corroborated with a checklist. Without seeing the checklist, I wonder how sure the authors can be about the designation. It would be good to know if scores on the checklist have been previously shown to correlate with self-report, or other more objective assessments, of dyslexia. Moreover, was there any overlap in scores between the dyslexic and non-dyslexic participants?

AUTHORS: Thank you for pointing out the misunderstanding. All dyslexic participants of our study had an official diagnosis of dyslexia prior to this research taking place. We clarified this on p. 10 (lines 487-488), to prevent such misunderstandings. In lines 497-498, we further added that there was no overlap in the dyslexia scores between the two groups. We also realised that our reference management software created an incorrect entry for the dyslexia checklist which we corrected (see entry 116, with a link to the checklist we used). As advised by the reviewer, we added an external reference that previously documented a strong correlation between the checklist scores and more objective measures of dyslexia (see entry 120, referred to in lines 492-494).

R1-C2: I’m also concerned about power issues with these data. Are 13 and 11 in the two groups enough to demonstrate significant differences between groups? There’s the additional factor of musical training. My concern about low power is most important in thinking about the lack of differences in the phoneme monitoring results. Minimally, I think the authors need to provide a power analysis to show that they could uncover differences between the groups in the phoneme monitoring task. But I also worry that there is enough variability within the population of dyslexic adults (mentioned in the current paper, in fact) that this sample size cannot capture true differences. For example, Leong & Goswami (2014) tested almost twice this number of participants.

AUTHORS: The reviewer is absolutely right in pointing out that the study by Leong & Goswami (2014) was based on the SMS data collected from 21/22 dyslexic/control participants while our study is based on the SMS data collected from 13/11 dyslexic/control participants (owing to the pandemic). However, Wolff (2002, cited in sect. 1.2 of the revised manuscript) tested SMS in a sample of a similar size (12 dyslexic adolescents). Similarly, Siok et al. (2009, Current Biology – not cited in the manuscript because the study is focused on Mandarin Chinese) tested 12 dyslexic and control participants on a large number of tasks. We added the requested post-hoc power analyses to the Methods sect. 2.5 (see lines 517-525). Alas, clinical (including dyslexic) data tend to show a lot of individual variation (Tallal, 1980; see Protopapas 2014 for an overview), and our dataset is no exception.

R1-C3: I think it would help for the authors to make explicit in the Method that the SMS task requires tapping in time with ALL syllables. I know this is pointed out in the Intro, but it would be helpful to be explicit in the Method as well.

AUTHORS: We added a statement to clarify this point to lines 435-438 as requested.

R1-C4: Line 394: “tap on the tambourine” The tambourine needs to be introduced before this mention. It’s introduced in the next section on the SMS task.

AUTHORS: Thank you for catching this, we corrected it by moving the relevant explanation from sect. 2.2 (lines ~ 447-449) to sect. 2.1 (lines ~400-402).

R1-C5: Line 540: There’s an indication of a footnote, but I can’t find the footnote being referenced?

AUTHORS: Corrected, thank you for spotting it.

R1-C6: Line 552: Remind the reader what group is referring to here.

AUTHORS: Added as advised.

R1-C7: Line 589 – what is the test here (z = 2.36, p > .05) describing? Not clear, and should it be ‘p < .05’? There are other tests reported in this paragraph where the actual test isn’t clear. Please clarify.

AUTHORS: Indeed, thank you for spotting the typo. We clarified our analyses in a new (short) sect. 2.7, and added more information in lines 603-604. We further added references for all R-packages we used.

 R1-C8: I would recommend the authors cite Tierney, et al., 2021 (https://www.sciencedirect.com/science/article/pii/S0022096521001144), who also show a connection between reading ability and memory for rhythm.

AUTHORS: We added a reference to the suggested work in the discussion (sect. 4.2, lines 849-850).

Reviewer 2 Report

This manuscript presents the results of an experiment testing rhythm perception, motor entrainment and phoneme processing in dyslexic and control adults. The introduction section is quite extensive and provide a detailed view of the literature on this topic. Similarly, the discussion section is well-developed too. The methods are sound and the results seem to support the conclusion. However, the results section lacks some critical information. Statistics are only partially presented, preventing a deep understanding of the present results, especially how much they support the conlusions. 

In the Results section, the authors only provide statistical results where significance was reached. For example lines 539, 559… no stats are provided. It is problematic that the non-significant results cannot be appreciated by the reader (neither in the text, nor in the figures and tables). Evidently, a non-significant result with a marginal p value and/or a strong-effect size do not lead to the same interpretation than one with a strong p value and a weak effect-size. The authors need to provide these results (with effect sizes) to allow the reader to fully appreciate how much the results support the conclusion. This could help making the paper more convincing.

Although focusing on hearing function only, this manuscript overall questions sensory theories of dyslexia. Sensory impairments in dyslexia have been demonstrated for other senses such as vision, and somatosensation (Lovegrove et al., Science 1980, Specific reading disability : differences in contrast sensitivity as a function of spatial frequency; Vilhena et al., Arq Bras Ophtalmol 2021, Magnocellular visual function in developmental dyslexia: deficit in frequency-doubling perimetry and ocular motor skills; Stoodley et al. Neurosci. Lett. 2000, Selective deficits of vibrotactile sensitivity in dyslexic readers; Laprevotte et al. Sc. Reports 2021, Movement detection thresholds reveal proprioceptive impairments in developmental dyslexia), senses that are both known to participate to speech perception. Given the overall conclusion regarding sensory theories of dyslexia and because dyslexia is associated to impairments in multisensory integration (Hahn et al., Neurosci. and Biobehav. Rev. 2014, Impairments of Multisensory Integration and Cross-Sensory Learning as Pathways to Dyslexia; Harrar et al. Curr. Biol. 2014, Multisensory integration and attention in developmental dyslexia) a more comprehensive presentation/discussion of the sensory literature is necessary.

In the methods, several times (for example lines 419-421), the authors refer to a recent work rather than explaining the procedures (Ratchke et al. 2021). In addition to referring to this publication, providing a simple and short description of the methodology would greatly help the reader.

Lines 441-442, can the authors detail how the SMS-rate is defined and calculated?

Lines 465-468, can the authors detail how the musical score is defined and calculated?

Author Response

AUTHORS: We would like to thank the Reviewer for the appreciation of our work, and for the critical engagement with our research that helped us improve the manuscript. We addressed the issues identified by the reviewer, and outlined the corresponding changes below.

R2-C1: In the Results section, the authors only provide statistical results where significance was reached. For example lines 539, 559… no stats are provided. It is problematic that the non-significant results cannot be appreciated by the reader (neither in the text, nor in the figures and tables). Evidently, a non-significant result with a marginal p value and/or a strong-effect size do not lead to the same interpretation than one with a strong p value and a weak effect-size. The authors need to provide these results (with effect sizes) to allow the reader to fully appreciate how much the results support the conclusion. This could help making the paper more convincing.

AUTHORS: We added the required information about the beta values and the relevant statistical outputs to sect. 3.1 (we created Table 3), sect. 3.3.

R2-C2: Although focusing on hearing function only, this manuscript overall questions sensory theories of dyslexia. Sensory impairments in dyslexia have been demonstrated for other senses such as vision, and somatosensation (Lovegrove et al., Science 1980, Specific reading disability : differences in contrast sensitivity as a function of spatial frequency; Vilhena et al., Arq Bras Ophtalmol 2021, Magnocellular visual function in developmental dyslexia: deficit in frequency-doubling perimetry and ocular motor skills; Stoodley et al. Neurosci. Lett. 2000, Selective deficits of vibrotactile sensitivity in dyslexic readers; Laprevotte et al. Sc. Reports 2021, Movement detection thresholds reveal proprioceptive impairments in developmental dyslexia), senses that are both known to participate to speech perception. Given the overall conclusion regarding sensory theories of dyslexia and because dyslexia is associated to impairments in multisensory integration (Hahn et al., Neurosci. and Biobehav. Rev. 2014, Impairments of Multisensory Integration and Cross-Sensory Learning as Pathways to Dyslexia; Harrar et al. Curr. Biol. 2014, Multisensory integration and attention in developmental dyslexia) a more comprehensive presentation/discussion of the sensory literature is necessary.

AUTHORS: We appreciate the reviewer’s suggestion of placing our research in a broader context of theoretical debates surrounding the causes of dyslexia. We feel less qualified to comment on senses other than audition, and hope that our (cautionary) concluding paragraph (see lines 992-1000) does at least some justice to the complexity of the topic, as the reviewer rightfully points out. We added all references that the reviewer suggested.

R2-C3: In the methods, several times (for example lines 419-421), the authors refer to a recent work rather than explaining the procedures (Rathcke et al. 2021). In addition to referring to this publication, providing a simple and short description of the methodology would greatly help the reader.

AUTHORS: We added more information about our methods to sect. 2.1 and 2.6. as requested.

R2-C4: Lines 441-442, can the authors detail how the SMS-rate is defined and calculated?

AUTHORS: We added the information to the end of sect. 2.2. (lines 456-457) as requested.

R2-C5: Lines 465-468, can the authors detail how the musical score is defined and calculated?

AUTHORS: We added the information to the top of sect. 2.4. (lines 478-485) as requested.

Round 2

Reviewer 2 Report

The main concern brought about in my previous evaluation was the lack of details in the reporting of statistics. Although this aspect of the manuscript has been slightly improved in the revised version, the authors did not provide the required information: p-values and effect sizes. For example, in Table 3, It is only written n.s. where the test does not reach significance. Line 794-795, it is only said that no significant interaction was found… This does not improve the understanding of the result (see my comment in the previous evaluation). Each time the authors report non-significant results that will later, in the paper, be used to conclude that some processes are intact whilst others (where comparisons reached significance) are impaired, I respectfully ask that the authors provide detailed statistics: exact p-values and effect-sizes.

Line 643-644, how do the authors define that a participant synchronized with a syllable and not with another? What methodology was used: time window, threshold…? Can the authors better explain this aspect to ensure that their analyses can be replicated in the future.

All other comments of my previous review have been addressed by the authors.

Author Response

The main concern brought about in my previous evaluation was the lack of details in the reporting of statistics. Although this aspect of the manuscript has been slightly improved in the revised version, the authors did not provide the required information: p-values and effect sizes. For example, in Table 3, It is only written n.s. where the test does not reach significance.

AUTHORS: We are not familiar with the way of statistical reporting requested by the reviewer and would appreciate if the editorial team could confirm that the implemented statistical reporting is in alignment with the journal’s guidelines. We added the numerical values to all non-significant p, and included Cohen’s d in Table 3.

Line 794-795, it is only said that no significant interaction was found. This does not improve the understanding of the result (see my comment in the previous evaluation). Each time the authors report non-significant results that will later, in the paper, be used to conclude that some processes are intact whilst others (where comparisons reached significance) are impaired, I respectfully ask that the authors provide detailed statistics: exact p-values and effect-sizes.

AUTHORS: Our line numberings seem to diverge here, as we cannot find which effect the reviewer is referring to, exactly. We added the information about all n.s. interactions of interest to all tables of the sections reporting the results, and to the main body of text where relevant.

Line 643-644, how do the authors define that a participant synchronized with a syllable and not with another? What methodology was used: time window, threshold…? Can the authors better explain this aspect to ensure that their analyses can be replicated in the future.

AUTHORS: We added the information requested, with a specified time-window and a reference to the example given in Figure 1.